# Collaborating with language models for embodied reasoning

**Ishita Dasgupta**[*]
DeepMind

**Christine Kaeser-Chen**
DeepMind

**Kenneth Marino**
DeepMind

**Arun Ahuja**
DeepMind

**Sheila Babayan**
DeepMind

**Felix Hill**
DeepMind

**Rob Fergus**
DeepMind

## Abstract

Reasoning in a complex and ambiguous environment is a key goal for Reinforcement Learning (RL) agents. While some sophisticated RL agents can successfully solve difficult tasks, they require a large amount of training data and often struggle to generalize to new unseen environments and new tasks. On the other hand, Large Scale Language Models (LSLMs) have exhibited strong reasoning ability and the ability to to adapt to new tasks through in-context learning. However, LSLMs do not inherently have the ability to interrogate or intervene on the environment. In this work, we investigate how to combine these complementary abilities in a single system consisting of three parts: a Planner, an Actor, and a Reporter. The Planner is a pre-trained language model that can issue commands to a simple embodied agent (the Actor), while the Reporter communicates with the Planner to inform its next command. We present a set of tasks that require reasoning, test this system's ability to generalize zero-shot and investigate failure cases, and demonstrate how components of this system can be trained with reinforcement-learning to improve performance.

## 1   Introduction.

Achieving complex tasks in embodied environments often requires logical reasoning. Such logical reasoning has been a challenge for machine learning [Russin et al., 2020, Mitchell, 2021] – even more so with embodied agents, where the agent also has to *perceive* and *control* in its environment, in addition to *reasoning* about how to accomplish a complex task. Recent large scale language models (LSLMs), however, have shown great promise for reasoning [Radford et al., 2019, Brown et al., 2020]. Can this complex reasoning ability be used for embodied tasks?

One major issue is that LSLMs are not embodied or grounded. They do not have a way to directly take actions in embodied environments, or of knowing what is happening in an environment. For each of these, we rely on other components of an agent model. In this work, we investigate an agent paradigm that we call **Planner-Actor-Reporter**. The **Planner** is the LSLM—it reads the task description, does any required logical reasoning, and breaks the problem down into a sequence of simple instructions. These instructions are passed to the **Actor**, which is an RL agent programmed to complete a small set of simple instructions in the environment. Finally, to complete the feedback loop, we have the **Reporter**, which observes the environment and reports information back to the Planner so it can adjust the instructions it issues. See Figure 1A.

Other recent work has investigated forms of closed-loop feedback for LSLMs in embodied reasoning tasks Huang et al. [2022], Ahn et al. [2022]. In this work, we generalize these approaches into a three part Planner-Actor-Reporter paradigm. We highlight the separate and crucial roles played

---

[*]corresponding author; idg@google.com

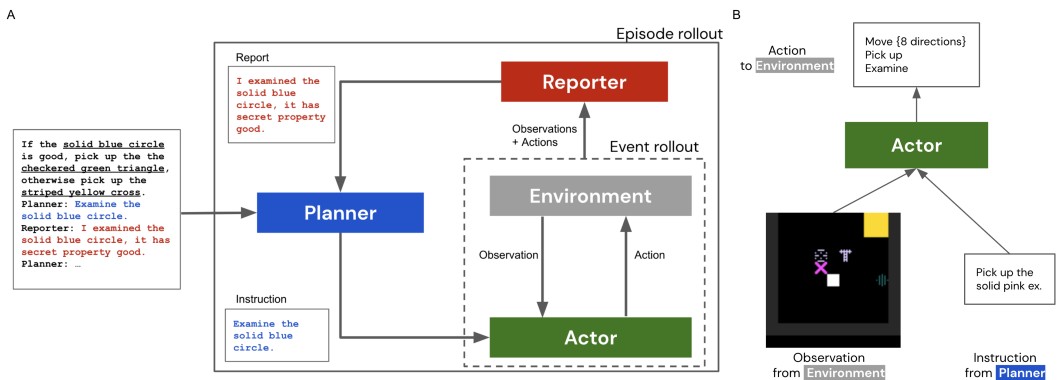

Figure 1: **Setup.** A. Schematic of the **Planner-Actor-Reporter** paradigm and an example of the interaction among them. B. Observation and action space of the PycoLab environment.

by these components by introducing and evaluating on a series of tasks which require the agent to explore the world to gather information necessary for planning, break down complex tasks into steps, and communicate visual properties of the world back to the Planner. Finally, we demonstrate that the Reporter module can be trained with reinforcement learning (RL), reducing the need for hand-specified sources of feedback.

## 2 Methods

**Environment and Actor:** Our environment is a 2D partially observable grid-world. The environment contains unique objects specified by color, shape and texture, and the Actor sees a top-down egocentric pixel RGB view with visibility within 5 squares of the agent. In addition to movement actions, the Actor can perform two special actions when on top of an object: *examine* which reveals a hidden piece of text about the object, and *pickup* which adds the object to its inventory.

The Actor is pre-trained with RL to follow instructions of the form "Pick up the X" or "Examine the X". Figure 1B shows an example observation from the environment, details about Actor architecture and environment can be found in App B.

**Planner:** We use pre-trained large language models with the same architecture: Chinchilla [Hoffmann et al., 2022], of two sizes: 7B and 70B parameters. To promote grounding with in-context learning [Brown et al., 2020], we provide 5 randomly selected "few-shot examples" of each task (assuming optimal Planner, Reporter, and Actor; see App E for full text), and directly use the model's sampled language as input to the Actor. At every timestep, the sampled language and information generated by the Reporter are appended to the dialogue transcript, and used as the prompt to get a new instruction from the Planner at the next timestep.

**Reporter:** We specify the role of the Reporter further by drawing parallels to hierarchical RL [Sutton et al., 1999, Kulkarni et al., 2016], where a high-level 'Planner' issues temporally abstracted instructions to a lower-level 'Actor'. A key difference from these setups is that in our experiments, the observation space of the Actor and Planner are different. In our setup, the Actor operates over pixel observations and produces movement actions, while the Planner operates over a language observation (the prompt) and produces language actions (the produced instruction). The Actor is language conditional and can interpret the Planner's instructions. But the Planner cannot parse the results of the Actor's actions (to produce an appropriate next action). The Reporter translates from the Actor's action+observation space to the Planner's. In the most general case, a Reporter takes (a sequence of) Actor actions and pixel observations and produces a text string that contains *true* and *relevant* information about what the Actor did and how the environment responded.

There are several ways to implement a Reporter, varying what is reported and how much of it is hard-coded, pre-trained, or learned from scratch. Previous work has used implicit Reporters implemented as part of the Actor that only convey instruction-completion [Ahn et al., 2022], or pre-trained perception models that answer natural language questions about the Actor's observations [Zeng et al., 2022, Huang et al., 2022]. In this work, we start with a hard-coded Reporter to first explore the performance of the Planner-Actor interaction in our novel information gathering tasks (Sec 3). We then pioneer learning this Reporter within the Planner-Actor-Learner loop to optimize reward (Sec 4).

**Tasks:**   We create a suite of tasks that examine the challenges of reasoning, generalization, and exploration in embodied environments that LM Planners can help with (detailed in App C). We focus on two types of tasks (*conditional* and *search* tasks) that require explicit information gathering such that a) the Planner must issue an explicit information gathering instruction, b) the Actor must carry it out, and c) the Reporter must relay the results before d) the Planner can issue the next instruction.

## 3   Language models as interactive planners

We examine the interaction between Planner, Actor and Reporter in tasks that require all three components for success. Building on top of previous work [Ahn et al., 2022, Zeng et al., 2022] which show that LSLMs can break down a complex real-world tasks into step-by-step instructions, we focus on tasks where the Planner needs to also explicitly issue information gathering instructions and incorporate the reported information for generating the next instructions. Further, our tasks are realized over objects with abstract properties that are not grounded in the LM's previous semantic experiences and therefore require significant abstract logical reasoning. We analyze the performance of different Planners and their robustness. All components are pre-trained.

The task setup is as follows: all the objects in the room have a 'secret property' (good / bad / unknown). When the Actor 'examines' an object, a ground-truth Reporter relays a text string 'I examined {object}, its secret property is {value}' to the Planner. The Planner can then issue the next instruction to the Actor.

### 3.1   Secret property conditional task

We start with the simplest task which requires information gathering. The goal of the episode is to pick up a correct object, based on another object's secret property. The task description passed to the Planner is as follows: 'If {decider object} is good, pick up {object 1}, otherwise pick up {object 2}'. A successful episode consists of 5 steps: a) the Planner instructs the Actor to examine the {decider object}, b) the Actor examines the object, c) the Reporter relays the revealed information (always done correctly in this setting), d) the Planner reasons which object needs to be picked up based on the report, {object 1} or {object 2}, and instructs the Actor to pick up the correct object e) the Actor picks up the correct object.

Explicit information gathering actions are classically challenging with pure RL. With a LSLM Planner and 5 language traces of solved examples as prompt, and an Actor trained on only simple pick-up and examine tasks, we can complete this complex multi-step task with good accuracy (Fig 2A). A pure RL baseline performs poorly even after 100M learner frames (see App D, Fig 2A).

In our analysis, we identify two main failure cases: the LSLM Planner failing to infer the next instruction given the environment feedback, and the Actor failing to follow the instruction provided by the Planner. In the first case, we observe that smaller language models (7B parameters) are only able to infer the correct object to pick up for reward 58% of the time given all information; larger language models (70B parameters) are able to do so 96% of the time. This shows that even relatively simple reasoning remains out of reach for smaller models without fine-tuning. In the second failure case, we observe that the Actor might encounter distribution shift, for example in episode length or instruction format, which makes it unable to Planner's instruction.

### 3.2   Secret property search task

We extend the previous task by requiring additional steps of information gathering. Instead of examining a single object, the agent needs to examine multiple objects, note their secret properties, and pick up the correct object for reward. The task is specified as 'The objects are {}, {}, {}, and {}. Pick up the object with the good secret property'. A successful episode consists of the Planner asking the Actor to examine each object in turn until it finds one with a 'good' property, at which point it asks the Actor to pick up that object.

Although this task requires more information gathering steps, and the RL baseline performs worse (see App D), the agent framework with Planner–Actor–Reporter is still able to complete the task zero-shot (i.e. without any additional environment interaction; Fig 2A). Curiously, we observe that our agents perform better in this task than in the previous task where only one object needs to be examined (Fig 2A; and App D). We hypothesize that since the number of information gathering steps varies, the Planner doesn't use a rigid "one examine, one pick up" policy and can be more robust to errors. For example, if the Actor examines the wrong object. We see that the Planner can indeed recover from such errors (Sec A.1). Similar to the observations above, we note that larger language models (70B) perform significantly better than smaller models (7B) (Fig 2A).

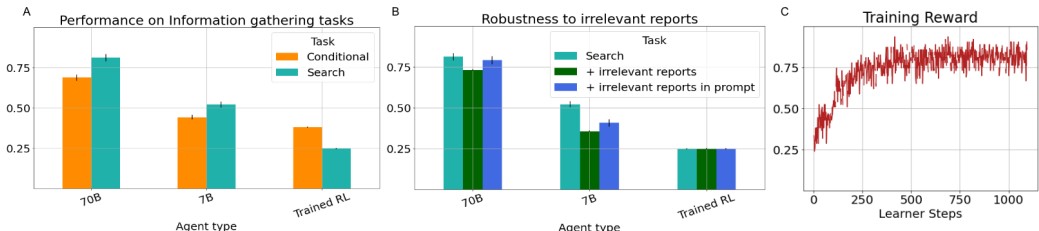

Figure 2: **Results.** A. Performance on secret property conditional and secret property search tasks with different Planners and baseline RL. B. Robustness of the Planners under an imperfect Reporter on the secret property search task. C. Improvement in performance as a Reporter is trained on the Visual conditional task. All error-bars are CIs across multiple episodes.

### 3.3 Robustness to irrelevant reports

We saw in the *search* task from the previous section, that the 70B Planner is reasonably robust to mistakes from the Actor (e.g. Section A.1). In this section, we examine if it can also be robust to a noisy Reporter. We break the assumption that only task relevant actions in the environment are reported, and irrelevant actions in the environment, e.g. "I have moved left" / "I have moved up and right" etc. are reported 20% of the time.

We find that performance does reduce but not dramatically (Fig 2B). The smaller 7B model is less robust than the 70B model, showing a more dramatic reduction in performance. We find that the 70B Planner uses strategies of repetition (where it repeats an instruction until it receives the relevant report, e.g. Sec A.2) or cycling (where it cycles through examine instructions for all the objects, e.g. Sec A.3), or some combination of the two, until it hits a 'good object'.

The few shot prompts provide no examples of how to respond to irrelevant reports. When we do provide guidance and demonstrate a 'repeating' strategy (e.g. Sec A.2) in the prompted examples, this restores performance to that without the irrelevant reports for the 70B Planner (Fig 2B); the 7B Planner improves but doesn't fully recover. This robustness indicates promise that our approach (particularly with large Planners) scales to imperfect Reporters. However, inference time through a large Planner is expensive, so a Reporter that ignores irrelevant events is more efficient.

## 4 Training a truthful Reporter

In the previous section, we focused on studying the behaviors of the Planner in our agent framework with a Reporter which always reports accurate information. However, such a Reporter does not exist in most environments. In this section, we study how we can train a reporter from scratch with RL.

We consider a 'visual conditional task' where the "secret property" is not directly revealed in text with a special 'examine' action, but rather must be decoded from visual observations. In particular, the task is specified as 'If {decider object} is close to the wall, pick up {object 1}, otherwise pick up {object 2}'. The Reporter's input is the same visual observations as the Actor and its output is a binary classifier head that can choose between one of two reports ('The object is {close to /far from} the wall'). Note that when training first starts, the Reporter does not have any pre-existing grounding mechanisms to report accurate information about the scene. As training continues, the Reporter can use the final reward of the episode to learn what information is most *helpful* to the Planner, and eventually converge to report only truthful and relevant information.

In contrast, recent work has used pretrained models with visual grounding (e.g. vision language models Zeng et al. [2022], or handcrafted mechanisms Huang et al. [2022]) to act as the Reporter module. We believe that building an effective Reporter module should combine both approaches: using a pre-trained module to bootstrap perception and grounding, and then using RL to finetune the pre-trained module to communicate with the Planner module. Our investigations show that Reporter training with RL is indeed viable and beneficial.

## 5 Discussion and future work

We advocate for a three-part system (Planner-Actor-Reporter), using pre-trained language models as a Planner that issues natural language commands to an embodied Actor, with a Reporter translating information back to the Planner. We introduce a series of tasks that leverage a pre-trained language model's abstract reasoning capacities, showing impressive and robust zero-shot performance, and analyse errors in different-sized models. We show the first proof of concept that the Reporter can

be trained to facilitate better collaboration between Planner and Actor. Exciting directions for future work include incorporating pre-trained components into the Reporter, expanding to more complex/realistic tasks, and improving training with a large model in the loop.

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

# A   Examples of Planner-Reporter-Actor dialogue

In this section, we include examples of Planner-Actor-Reporter dialogue, in particular for the secret property search tasks, demonstrating Planner robustness to Actor and Reporter errors.

## A.1   A more robust search policy.

The red box shows where the Actor made a mistake, the green box highlights the Planner's recovery.

```
QUESTION: The objects in the room are vertical striped red inverse ex, horizontal striped teal circle, vertical
striped dark blue tee and vertical striped pink inverse ex. Get the object with a good secret property.
ANSWER:
LM: Examine vertical striped red inverse ex.<EOS>
Agent: I examined vertical striped pink inverse ex. Its secret property has value bad.<EOS>
LM: Examine horizontal striped teal circle.<EOS>
Agent: I examined horizontal striped teal circle. Its secret property has value bad.<EOS>
LM: Examine vertical striped dark blue tee.<EOS>
Agent: I examined vertical striped dark blue tee. Its secret property has value bad.<EOS>
LM: Examine vertical striped pink inverse ex.<EOS>
Agent: I examined vertical striped pink inverse ex. Its secret property has value bad.<EOS>
LM: Examine vertical striped red inverse ex.<EOS>
Agent: I examined vertical striped red inverse ex. Its secret property has value good.<EOS>
LM: Pickup vertical striped red inverse ex.<EOS>
```

## A.2   Emergent strategies in response to irrelevant reports: Repeating strategy.

The Planner repeats commands until it is completed, even in response to irrelevant reports.

```
QUESTION: The objects in the room are noisy dark red tee, vertical striped cyan plus, noisy purple u and horizontal
striped pink triangle. Get the object with  a good secret property.
ANSWER:
LM: Examine noisy dark red tee.<EOS>
Agent: I have moved up and right.<EOS>
LM: Examine noisy dark red tee.<EOS>
Agent: I have moved up and right.<EOS>
LM: Examine noisy dark red tee.<EOS>
Agent: I have moved down and right.<EOS>
LM: Examine noisy dark red tee.<EOS>
Agent: I have moved down and left.<EOS>
LM: Examine noisy dark red tee.<EOS>
Agent: I examined noisy dark red tee. Its secret property has value bad.<EOS>
LM: Examine vertical striped cyan plus.<EOS>
Agent: I have moved left.<EOS>
LM: Examine vertical striped cyan plus.<EOS>
Agent: I examined vertical striped cyan plus. Its secret property has value good.<EOS>
LM: Pickup vertical striped cyan plus.<EOS>
```

## A.3   Emergent strategies in response to irrelevant reports: Cycling strategy.

The Planner cycles through examine commands until it receives a report of a good object.//

```
QUESTION: The objects in the room are grid pink tee, horizontal striped yellow upside down tee, vertical striped
light yellow h and noisy dark blue upside down u. Get the object with a good secret property.
ANSWER:
LM: Examine grid pink tee.<EOS>
Agent: I have moved up.<EOS>
LM: Examine horizontal striped yellow upside down tee.<EOS>
Agent: I have moved up and right.<EOS>
LM: Examine vertical striped light yellow h.<EOS>
Agent: I have moved down and left.<EOS>
LM: Examine noisy dark blue upside down u.<EOS>
Agent: I examined noisy dark blue upside down u. Its secret property has value bad.<EOS>
LM: Examine grid pink tee.<EOS>
Agent: I have moved down and left.<EOS>
LM: Examine horizontal striped yellow upside down tee.<EOS>
Agent: I have moved down.<EOS>
LM: Examine vertical striped light yellow h.<EOS>
Agent: I have moved down.<EOS>
LM: Examine noisy dark blue upside down u.<EOS>
Agent: I examined vertical striped light yellow h. Its secret property has value bad.<EOS>
LM: Examine grid pink tee.<EOS>
Agent: I have moved up and right.<EOS>
LM: Examine horizontal striped yellow upside down tee.<EOS>
Agent: I examined grid pink tee. Its secret property has value good.<EOS>
LM: Pickup grid pink tee.<EOS>
```

# B  Environment and agent architecture details

## B.1  Grid-word environment

We implemented a grid-world environment with the PyColab library (https://github.com/deepmind/pycolab). The grid-world is 11x11, with the outer border being impassable walls, and no internal obstacles. There are four objects and one agent in the environment, each occupying one grid. Their start locations in each game are randomly assigned when the episode starts.

We assign objects in the environment randomly selected *color*, *texture*, and *shape* attributes, from pre-defined lists of allowed values. The exact object attribute combinations in evaluation tasks are held out from the training tasks, though each individual attribute value has been observed in training.

The agent's view on the environment is always a 11x11 crop of the scene from a top-down perspective, with the agent at the center. As the agent moves around, the crop scrolls to keep the agent centered in the visual observation.

## B.2  Agent architecture

Our agent comprises three modules: a Planner, an Actor, and a Reporter. The Planner module is a pretrained LSLM. In our experiments, we use two variants of the Chinchilla models [Hoffmann et al., 2022], the one with 70 billion parameters (referred as the 70B model), and the one with 7 billion parameters (referred a the 7B model).

The Actor module uses a pre-trained policy, trained on simple tasks in the same grid-world environment. The policy uses a simple convolutional visual encoder with 3 layers to encode visual observations, and a LSTM-based language encoder to encode action instructions. The agent also has a LSTM-based memory module to help take previous actions and observations into account for policy output. The policy head for the Actor outputs a distribution in the action space, which in our case contains the discrete movement actions (e.g. move up or down in the grid-world), and the special actions of *pick up* and *examine*. The actual agent action is then sampled from the policy output.

The Reporter module shares a lot of similarities with the Actor architecture: it also comprises visual and instruction encoders, a memory module, and a policy head. In the experiments described in Chapter 4, the policy head is simply a binary distribution over two pre-defined text reports. Though in future work we do plan to allow direct language generation from the Reporter module.

### B.3  Training procedures

Both the Actor and Reporter module are trained with standard VTrace loss [Espeholt et al., 2018]. The Actor training converged after about 100,000 frames, while the Reporter training converged after 500 frames.

We caution that despite the number of frames needed for Reporter training is not many, the training time can be very long. This is because in each episode, the LSLM Planner may need to be queried several times. Inference time of these models is quite slow still, which increases the amount of wall time needed to collect enough trajectories to train the agent.

## C  All task descriptions

We frame the capabilities of our agent, and similarly, the requirements of our task suite around four fundamental aspects of embodied intelligence:

1. **Logical Reasoning:** The ability to take complex instructions and do different kinds of logical operations on them to determine the correct course of action.

2. **Generalization:** The ability to generalize beyond the agent's previous experience.

3. **Exploration:** The ability to explore the world around the agent to uncover new information that can inform its reasoning for what actions to take.

4. **Perception:** The ability to use the raw observation the agent has (usually vision) and process the world and use what it sees to make decisions.

**Logical Reasoning:**    *The ability to take complex instructions and do different kinds of logical operations on them to determine the correct course of action.*

This ability applies mainly to an embodied agent's ability to interpret the meaning of a goal specification (prompt) as well as integrate other information it has about the environment. This can include if-else conditionals, choosing from among options by eliminating options, choosing objects that match certain properties, etc. LSLMs are particularly adept at these kinds of logical language tasks [Rae et al., 2021, Hoffmann et al., 2022].

**Generalization:**    *The ability to generalize beyond the agent's previous experience.*

The ability to generalize to new inputs has been well studied in RL [Kirk et al., 2021], but remains a significant challenge. In the field of LSLMs, we have seen remarkable success in few-shot [Brown et al., 2020] learning to new text tasks and inputs. These 'few-shots' are language traces of optimal behavior provided in the prompt, the agent never receives new interaction data or demonstrations for any of the generalization tasks. Language descriptions are much cheaper and easier to collect for new tasks than demonstrations or interactions. We examine generalization from the 'train' examples provided in the prompt to a new test prompt.

We study generalization of **objects**, where tasks have been seen in training with a specific set of objects and tested with other objects. We study generalization in language **prompts**, where the same task can be communicated in several different ways, but the structure of the task itself remains the same. Finally, we study generalization in the **task** itself, e.g. collecting 3 objects when the train traces only showed 2.

**Exploration:**    *The ability to explore the world around you to learn new information that can inform your reasoning for what actions to take.*

This definition is subtly different from the way exploration is often used in the RL literature. Here we do not mean the ability during training to explore the policy space to find a more optimal policy (versus exploiting a known policy). We mean the agent's ability to actively discover information that is not observable in its current state. In our tasks, agents must move to uncover textual clues about objects using the "examine" action, or use perception to look at objects not currently in the agent's view.

**Perception:**    *The ability to use the raw observation the agent has (usually vision) and process the world and use what it sees to make decisions.*

This is ultimately the most fundamental agent ability, to observe its surroundings and to make sense of them. It is fundamental to RL in such a way that it is often not even mentioned as a core capability. This is however a core shortcoming of LSLMs, that operate purely over text and are not grounded. A core contribution of this work is to show how we can combine these complementary capabilities.

## C.1   Tasks

Finally, we present a series of tasks that exemplify the challenges discussed above.

**Option Elimination:**   Here we look at logical reasoning and (object and prompt) generalization. Consider the following motivating example:
`I know the corkscrew is either in the cutlery drawer or on the wine cabinet.`
`I just checked the cutlery drawer, but it wasn't there.  Can you find me`
`the corkscrew?`
The intended behavior from the agent is relatively simple, i.e. to "go to the wine cabinet". However, it is non-trivial to infer this simple instruction, and we examine if LSLMs can help. We design a task in our PyColab environment to emulate this kind of task. At the beginning of every episode, we select 4 unique objects and place them randomly. We then generate a templated instruction string with the names of these objects. We use 10 different language formats, we choose the Nshot prompts from 7 of them, and hold out 3 as test formats to examine generalization across prompts. In all of these, 3 of the 4 objects are "eliminated" as the target object, and optimal behavior is for the agent to go to the final object that was not eliminated. If the agent interacts with any other object, the episode ends without reward.

**Basic Step Tasks:**   We consider the simplest example of this where the task is to pick up two objects in the room e.g. "Pick up object X and Y in that order." The hope is that the language model can break this instruction down into its components, i.e. "Pick up X" and "Pick up Y" and issue them to the low-level agent in that order. This is the first multi-step task and therefore needs a Reporter (to tell the Planner when to issue the next instruction). We assume a truthful-and-relevant reporter that produces a text string "I have picked up <object name>." whenever the agent picks up an object. We examine generalization to different numbers of objects.

**Conditional Secret Property:**   Consider the following example: `Get me the coffee if it is still warm, otherwise get me a soda`. The language model has no way to know whether the coffee is warm without the help of an embodied agent examining the coffee and reporting back on its temperature, based on which the language model can then issue the next instruction. To emulate this kind of information in our gridworld – like temperature or texture that can only be gathered by an embodied agent with direct interaction with an object – we assume that each object has a list of hidden properties. When the agent does a special "examine" action on an object, the environment produces a text string listing all of these properties.

We start with 4 randomly placed unique objects in the room, and the agent's goal is to pick up a single target object. The challenge come from figuring out what the right target object is – this requires information gathering interaction with the environment. In our task, the agent has to examine a single arbitrarily assigned decider object that might have the property *good*, *bad*. Depending on which it is, the agent has to pick up a different target object. Objects apart from the decider object are not good or bad and instead of secret property unknown, examining an non-decider object X returns the string 'I examined X. Its secret property has value unknown.'.

**Search Secret Property:**

Consider the following example: `Can you bring me my water bottle?  It's either in the fridge, in my gym bag, or in my backpack`. The LM doesn't know where the water bottle is, the low-agent has to collect this information. The challenge in this task is to form and issue sequences of information gathering actions, recognize when the agent has received the necessary information to stop further information gathering, and issue a final instruction based on the results of this information gathering.

We emulate this structure in a search task where one of the four objects in the room is the target, but the agent does not know which one. This information is in the hidden properties of the objects: three of the four objects have the property *bad*, the remaining object has the property *good*, and the agent has to pick up the *good* object. Examining an object X returns the text string 'I examined X. Its secret

property has value good/bad.' The agent therefore has to sequentially examine each of the objects; when it encounters a *good* object, it should pick it up.

**Visual Color Conditional:**

We start with a simple conditional task, where the target object changes depending on the color of the agent. This agent color information is not available to the language model. A reporter much therefore learn to decode it from visual observations, and report it back. The language model can then issue the right next instruction, which leads to reward. This task doesn't require information gathering actions or commands, as the agent color is directly visible to the agent when it spawns.

**Visual Location Conditional:** The previous task was effectively a single step task – the information relevant to knowing what the target is (this information being the color of the agent), is available right from the first step of the task. To combine the challenges of multistep tasks examined in previous section with learning to report, we consider a conditional task more similar to the one examined previously, where the agent has to do an information gathering action in order to get the relevant information. We consider a conditional task in which which object the agent must pick up depends on whether a designated decider object is close to the wall. In order to gauge this, the agent has to navigate to the object to examine its surroundings.

# D   RL Baselines

We also trained the Actor (pre-trained on the 'pick up ' and 'examine' tasks) directly on the secret property conditional and search tasks (3). Note that we see a learning curve for the basic 'pick up ' and 'examine' tasks as well because of a slight distribution shift — the Actor here additionally receives the event report as a separate observation (encoded with the same architecture used to encode the instruction), this is necessary to enable to perform the information gathering tasks (i.e. to use the information generated). For the the basic pick up and examine tasks, the agent just has to learn to ignore this field (giving fast learning).

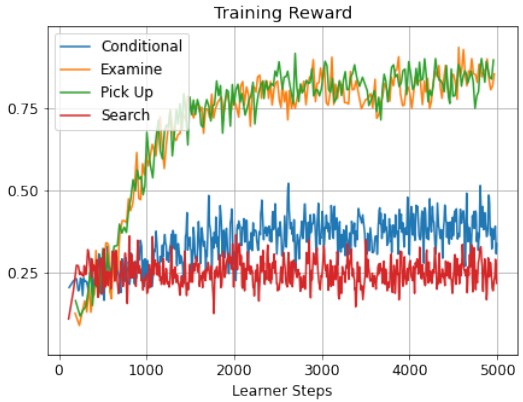

Figure 3: **Pure RL baselines.** See main text for details.

We find that after 5000 learner updates (140M learner frames), performance on the Search task is around 25% (chance performance when picking up a random object, without first examining it to determine if it is good), performance on the Conditional task is slightly higher, around 33% reflecting that the RL agent can learn to pick up one of the three objects in the prompt (and ignore the fourth), but doesn't learn to first examine the decider object to figure out which one of the target objects to pick up.

These baselines are not intended to convey that these tasks are impossible to learn with pure RL – better architectures discounting, exploration strategies etc will no doubt significantly improve performance. However, we demonstarte that even with several environment interactions, a pure RL baseline does find these tasks difficult – while our Planner-Actor-Reporter agent can perform fairly well with no additional environment interactions, and only 5 examples of optimal performance in the Planner's prompt.

# E All prompt formats used

## E.1 Conditional task prompt

```
QUESTION: If the solid dark blue h is good, pickup horizontal striped light
green inverse plus. Otherwise, pickup checker brown tee.
ANSWER:
LM:
Examine solid dark blue h.<EOS>
Agent:
I examined solid dark blue h. Its secret property has value bad.<EOS>
LM:
Pickup checker brown tee.<EOS>
Agent:
I picked up checker brown tee.<EOS>
DONE

QUESTION: If the noisy light green plus is good, pickup solid orange circle.
Otherwise, pickup vertical striped blue upside down u.
ANSWER:
LM:
Examine noisy light green plus.<EOS>
Agent:
I examined noisy light green plus. Its secret property has value bad.<EOS>
LM:
Pickup vertical striped blue upside down u.<EOS>
Agent:
I picked up vertical striped blue upside down u.<EOS>
DONE

QUESTION: If the vertical striped lavender upside down tee is good, pickup
solid purple inverse plus. Otherwise, pickup vertical striped green h.
ANSWER:
LM:
Examine vertical striped lavender upside down tee.<EOS>
Agent:
I examined vertical striped lavender upside down tee. Its secret property
has value bad.<EOS>
LM:
Pickup vertical striped green h.<EOS>
Agent:
I picked up vertical striped green h.<EOS>
DONE

QUESTION: If the vertical striped pink plus is good, pickup horizontal
striped blue tee. Otherwise, pickup vertical striped orange ex.
ANSWER:
LM:
Examine vertical striped pink plus.<EOS>
Agent:
I examined vertical striped pink plus. Its secret property has value
bad.<EOS>
LM:
```

```
Pickup vertical striped orange ex.<EOS>
Agent:
I picked up vertical striped orange ex.<EOS>
DONE

QUESTION: If the grid teal h is good, pickup vertical striped purple tee.
Otherwise, pickup vertical striped dark red upside down u.
ANSWER:
LM:
Examine grid teal h.<EOS>
Agent:
I examined grid teal h. Its secret property has value good.<EOS>
LM:
Pickup vertical striped purple tee.<EOS>
Agent:
I picked up vertical striped purple tee.<EOS>
DONE
```

## E.2   Search task prompt

```
QUESTION: The objects in the room are checker brown tee, horizontal striped
light green inverse plus, solid dark blue h and vertical striped blue tee.
Get the object with a good secret property.
ANSWER:
LM:
Examine checker brown tee.<EOS>
Agent:
I examined checker brown tee. Its secret property has value bad.<EOS>
LM:
Examine horizontal striped light green inverse plus.<EOS>
Agent:
I examined horizontal striped light green inverse plus. Its secret property
has value bad.<EOS>
LM:
Examine solid dark blue h.<EOS>
Agent:
I examined solid dark blue h. Its secret property has value good.<EOS>
LM:
Pickup solid dark blue h.<EOS>
Agent:
I picked up solid dark blue h.<EOS>
DONE

QUESTION: The objects in the room are noisy green plus, horizontal striped
blue u, noisy green circle and solid light green upside down u. Get the
object with a good secret property.
ANSWER:
LM:
Examine noisy green plus.<EOS>
Agent:
I examined noisy green plus. Its secret property has value bad.<EOS>
LM:
Examine horizontal striped blue u.<EOS>
Agent:
```

I examined horizontal striped blue u. Its secret property has value
bad.<EOS>
LM:
Examine noisy green circle.<EOS>
Agent:
I examined noisy green circle. Its secret property has value good.<EOS>
LM:
Pickup noisy green circle.<EOS>
Agent:
I picked up noisy green circle.<EOS>
DONE

QUESTION: The objects in the room are solid green triangle, solid yellow h,
checker lavender plus and noisy blue tee. Get the object with a good secret
property.
ANSWER:
LM:
Examine solid green triangle.<EOS>
Agent:
I examined solid green triangle. Its secret property has value good.<EOS>
LM:
Pickup solid green triangle.<EOS>
Agent:
I picked up solid green triangle.<EOS>
DONE

QUESTION: The objects in the room are horizontal striped brown tee, grid
yellow ex, solid dark red plus and checker light yellow ex. Get the object
with a good secret property.
ANSWER:
LM:
Examine horizontal striped brown tee.<EOS>
Agent:
I examined horizontal striped brown tee. Its secret property has value
good.<EOS>
LM:
Pickup horizontal striped brown tee.<EOS>
Agent:
I picked up horizontal striped brown tee.<EOS>
DONE

QUESTION: The objects in the room are vertical striped yellow plus, checker
dark red u, checker blue plus and vertical striped peach upside down tee.
Get the object with a good secret property.
ANSWER:
LM:
Examine vertical striped yellow plus.<EOS>
Agent:
I examined vertical striped yellow plus. Its secret property has value
good.<EOS>
LM:
Pickup vertical striped yellow plus.<EOS>
Agent:

I picked up vertical striped yellow plus.<EOS>
DONE}

