# OpenReview forum: "Collaborating with language models for embodied reasoning"
_NeurIPS.cc/2022/Workshop/LaReL — LaReL 2022_

### Official Review · Reviewer_jdBm · 2022-10-03
**Review for Collaborating with language models for embodied reasoning**

**Rating:** 8
**Confidence:** 5

**Review:**

This is a nice paper formalizing the idea of recent successes merging LMs and RL. It’s very well suited for this workshop, well written and clear.

I only have minor comments:

This paper is related to the idea of AlfWorld [1], a merge between the TextWorld and Alfred environment that combines a low-level policy pretrained in the sensorimotor world of Alfred to achieve simple instructions and a high-level policy pretrained in a TextWorld aligned to the Alfred world. Just like an LM, the high-level policy receives textual descriptions and gives back further instructions. I think in this paper, the Reporter is also hard-coded. To my knowledge, this is the first instance of aligned linguistic and sensorimotor space where the two types of decision making are combined.

I agree that the reporter needs to be trained. The challenge of training a classifier with two classes is very far from the objective of having an open-ended trajectory descriptor trained to describe relevant properties of the world/trajectory in light of the long-term instruction. How to move from one to the other is not totally clear here, but again, I acknowledge that it’s a difficult problem.

You could make a much wider argument for the use of LM as high-level planners. LMs are not only good for task decomposition (Ahn et al) and reasoning (this paper); they are especially good at capturing aspects of human culture. With LMs, agents could power new capabilities: exploration biases (search for knives in kitchens, not in the bedroom), theory of mind (A wanted a knife, he’s probably in the kitchen looking for it), social interactions (say hello and be polite when interacting with people), etc. Most of these things could not be learned by a standalone RL agent — they are cultural and need to be learned with/from humans, eg via an LM. See a more complete argument in that direction in [2].

* [1] Shridhar et al., 2020 – ALFWorld: Aligning Text and Embodied Environments for Interactive Learning
* [2] Colas et al., 2022 – Vygotskian Autotelic Artificial Intelligence: Language and Culture Internalization for Human-Like AI


Typos:
* L6: “to to adapt” (in the abstract)
* L12-15: “We present a set of tasks that require reasoning, test this system’s ability to generalize zero-shot and, investigate failure cases, and demonstrate how components of this system can be trained with reinforcement-learning to improve performance.”
* L33, use \citep instead of \cite here
* L113: “which makes it unable to Planner’s instruction” → to fulfill the Planner’s instructions?
* Please make the text bigger in Figure 1.
* Fig 2: CI is a function of the confidence level alpha, what’s the alpha ?

---

### Official Review · Reviewer_uDBN · 2022-10-14
**Review of Collaborating with language models for embodied reasoning**

**Rating:** 7
**Confidence:** 4

**Review:**

In this work, the authors argue that the abilities of Large-Scale Language-Models (LSLMs) to perform complex linguistic reasoning can be applied to embodied tasks. To provide necessary grounding to LSLMs, the authors introduce the Planner-Actor-Reporter paradigm. There the Planner, instantiated by a LSLM, uses previous task demonstrations as well as reports from the Reporter describing the previous low-level actions of the agent to propose the next task; the Actor, a pretrained language-conditioned policy, uses the instructions of the planner to act, and the Reporter converts these actions back into text feedback for the Planner. The authors demonstrate zero-shot performance of this system and investigate failure cases.

This work continues the previous line of research of applying pretrained LSLMs to embodied tasks by leveraging their reasoning capacities; studying how model size affects performance and recovery (7B param Chinchilla vs 70B) is an important demonstration of the importance of the emergent capacities of LSLMs and is appreciated.

The paper could stress how it differs from SayCan and similar works where there is also a LM Planner component and the selected skill is then appended to the context of the agent in a similar fashion. The major difference is the information-gathering nature of the task, and there the Reporter model is necessary to provide this information back to the Planner.

The tasks examined need all three of the proposed components to work well, and failures of the Planner were addressed (by increasing model size) as well as failures of the Actor (by showing examples of recovery by the Planner). However, in more complex setups, where the Reporter could be a pretrained image-to-text model describing complex scenes or events there is no guarantee that the Reporter would be faithful. I would thus like, in an extended version, to see the robustness of the method to unfaithful Reporters discussed or studied.

Overall, the paper is interesting; the framework seems general and applicable to other tasks, with other pretrained components. Relation to other very related methods should be discussed and the pertinence of this particular set of tasks could be stressed; a link to other similar approaches of turning environments into texual form such as [Alfworld ](https://arxiv.org/abs/2010.03768) could be made. I would like to see this paper at the workshop and think it will spark fruitful discusssions.

---

### Decision · Program_Chairs · 2022-10-21

Accept